# Selected Plant Triterpenoids and Their Derivatives as Antiviral Agents

**DOI:** 10.3390/molecules28237718

**Published:** 2023-11-22

**Authors:** Martina Wimmerová, Uladzimir Bildziukevich, Zdeněk Wimmer

**Affiliations:** 1Department of Chemistry of Natural Compounds, University of Chemistry and Technology in Prague, Technická 5, 16028 Prague, Czech Republic; wimmerom@vscht.cz; 2Isotope Laboratory, Institute of Experimental Botany, Academy of Sciences of the Czech Republic, Vídeňská 1083, 14220 Prague, Czech Republic; bildziukevich@biomed.cas.cz

**Keywords:** plant triterpenoid, structure modifier, antiviral activity, HIV-1, HSV-1, maturation inhibitor

## Abstract

The results of the most recent investigation of triterpenoid-based antiviral agents namely in the HIV-1 and HSV-1 treatment were reviewed and summarized. Several key historical achievements are included to stress consequences and continuity in this research. Most of the agents studied belong to a series of compounds derived from betulin or betulinic acid, and their synthetic derivative is called bevirimat. A termination of clinical trials of bevirimat in Phase IIb initiated a search for more successful compounds partly derived from bevirimat or designed independently of bevirimat structure. Surprisingly, a majority of bevirimat mimics are derivatives of betulinic acid, while other plant triterpenoids, such as ursolic acid, oleanolic acid, glycyrrhetinic acid, or other miscellaneous triterpenoids, are relatively rarely involved in a search for a novel antiviral agent. Therefore, this review article is divided into three parts based on the leading triterpenoid core structure.

## 1. Introduction

Currently, viral infections represent the main infectious disease worldwide [1,2], and they represent more than 65% of total number of infectious diseases [3]. Generally, viruses capable of invading humans belong into two categories: (a) viruses being long-term parasites in the human body (chickenpox, rubella, herpes, measles, smallpox, polio, Japanese encephalitis, mumps, cytomegalovirus, hepatitis B/C virus, dengue virus, human influenza virus, human immunodeficiency virus, human papillomavirus, etc.), and (b) viruses being long-term parasites in animals usually living close to humans (chickens, dogs, pigs, horses, sheep, etc.). These viruses are capable of infecting humans across species [4].

Viruses spread relatively easily in the human population. Over the past decade, the indisputable evidence of this effect of virus action has been documented by a wide transmission of fatal diseases that include severe acute respiratory syndrome (SARS), Middle East respiratory syndrome (MERS), Ebola, and other dangerous viruses having been considered as severe threats to human health. At present, vaccines and screening antiviral drugs represent the main way used for the prevention and treatment of human viral infections [5]. However, it is difficult to develop an effective vaccine because the side effects that have often been detected, have been unpredictable [6], and vaccination has not been effective in 100% of patients [7,8]. The main focus of antiviral drug research is limited to several types of viruses, such as HIV, herpes (HSV), influenza, hepatitis, and respiratory viruses [9]. Namely, anti-HIV-1 and anti-HSV-1 agents have been focused in more details.

The most effective anti-HIV-1 agents are compounds capable of inhibiting HIV-1 maturation [10]. Genetic and enzymatic investigation has resulted in a finding that inhibition of cleavage or even slowing cleavage at the CA-SP1 site should be sufficient to disrupt the maturation process significantly and to destroy virus infectivity quite efficiently. Therefore, maturation inhibitors interfering with CA-SP1 processing, have been the most important candidates for augmenting the current ways of treatment of infection by HIV [11,12]. Biochemical and structural studies revealed that a slow cleavage of CA-SP1 is due to the structural separation of the proteolysis site [13]. The detailed mechanism of inhibition has not yet been determined, however, small-molecule maturation inhibitors, such as 3-*O*-(3′,3′-dimethylsuccinyl)betulinic acid (bevirimat) and its analogs, have been supposed to interfere with proteolysis by binding to the CA-SP1 junction and stabilizing the 6-helix bundle [14,15,16,17]. Maturation inhibitors do not interfere with substrate binding directly, they rather act indirectly by inhibiting the unfolding of the 6-helix bundle resulting in preventing access of the protease to its substrate. Despite being potent inhibitors of HIV infection under laboratory conditions, maturation inhibitors have not yet been approved for clinical use. Bevirimat underwent Phase I and Phase II clinical trials, during which significant, dose-dependent viral load reductions in HIV-1-infected individuals were observed [18]. However, further studies revealed that in up to 50% of patients, bevirimat did not affect viral loads [19,20]. This resistance of bevirimat is clearly associated with naturally occurring viral sequence polymorphs.

HSV is a double-stranded DNA virus [21]. It exists in two types of HSV strains: HSV-1 causing infection of the oral mucosa, and HSV-2 affecting the genital mucosa and causing neonatal infections [22]. According to the latest report of the WHO, more than 65% of the world population under the age of 50 are infected with HSV-1, and the incidence of herpes simplex virus encephalitis (HSE) caused by HSV-1 infection is increasing every year. Moreover, HSV-1 infections are associated with neurodegenerative diseases, e.g., Alzheimer’s disease [23,24,25]. Intracranial infection caused by HSV-1 was identified as an important factor in the pathogenesis of Alzheimer’s disease [25], and the brain changes in some patients with herpes simplex virus encephalitis have been found to be similar to that of Alzheimer’s disease patients [25]. HSV-1 has also been associated with certain types of cancers, e.g., cervical carcinoma or acute lymphocytic leukemia [26]. At present, anti-HSV-1 drugs mainly belong among nucleoside drugs (e.g., acyclovir, ganciclovir, valacyclovir, etc.), capable of inhibiting viral replication by interfering with viral DNA polymerase. Based on the wide application of these drugs, drug-resistant strains of virus have appeared within a relatively short time period [25]. In order to cope with the problem of drug resistance, investigation has been focused on developing novel anti-HSV agents targeting the viral DNA polymerase. The HSV-1 life cycle includes adsorption and entry into the host cell, intracellular transport to the nucleus, DNA replication, gene transcription, protein synthesis, nucleocapsid assembly, and viral release [27]. Drugs theoretically capable of affecting any stage in the virus life cycle could be inhibitory [27].

## 2. Triterpenoid-Based Agents for Treating HIV-1 and HSV-1)

### 2.1. Plant Triterpenoids of the Lupane Family

Betulin (**1**; Figure 1), a pentacyclic natural triterpenoid, represents one of the potent and for a long time known plant-derived products [28]. Mostly, it has been found in the bark of various birch (*Betula*) species and can be extracted therefrom. The biological effects of betulin (**1**) have been intensively investigated, and have resulted in discovering its wide-ranging biological activity that involves antiviral, antibacterial, anticancer, and anti-inflammatory effects [29]. The antiviral properties of betulin (**1**) and its derivatives have been explored in the context of many different viruses [30,31,32]. Betulinic acid (**2**; Figure 1) represents another plant product extractable from the bark of birch and from other plant sources [33]. It can also be prepared synthetically by oxidation of betulin (**1**) [34]. The pharmacological characteristics of betulinic acid (**2**) are similar to betulin (**1**) [35,36,37]. Bevirimat, 3-*O*-(3′,3′-dimethylsuccinyl)betulinic acid (**3**; Figure 1), was one of the most promising derivatives of betulinic acid (**2**). It has displayed potent anti-HIV activity with a novel mechanism of action [38]. Unfortunately, its further development was terminated at Phase IIb of clinical trials due to the reduced efficacy of the compound against certain HIV strains [39]. Nevertheless, the discovery of bevirimat (**3**) initiated a subsequent investigation in the field of plant triterpenoids and their derivatives bearing various functional groups or structural modifiers, resulting in a synthesis of several early but successful bevirimat analogs (**4**–**8**; Figure 1) [40,41,42]. However, currently, approved antiviral drugs have very diverse structures [28,43].

Bevirimat (**3**) was the first-in-class HIV-1 maturation inhibitor. It showed a low efficacy, essentially due to the natural polymorphism of its target, the CA-SP1 junction. Moreover, its low solubility in water and in the physiological environment makes it difficult to study its interaction with the CA-SP1 junction. Therefore, designing new derivatives of bevirimat (**3**) was performed by introducing different hydrophilic substituents at the C-28 carboxyl group to improve the solubility of the novel compounds in aqueous media. A synthesis of the novel derivatives, the effect of substituents at the C-28 carboxyl group, and their solubility in aqueous media were investigated intensively, and the ability of these molecules to inhibit viral infection and their cytotoxicity was carefully evaluated [43]. Compared to the well-known bevirimat (**3**), one of the prepared compounds (**7**) showed higher solubility in aqueous media associated with a 2.5-fold increase in activity, higher selectivity index, and a better antiviral profile (Table 1) [43]. Moreover, for the first time, a direct interaction between the prepared compound (**7**) and the domain CA-SP1 was shown by the NMR (nuclear magnetic resonance) study [43]. Bevirimat (**3**) was launched by several pharmaceutical companies for further development and commercialization [44,45]. However, while bevirimat (**3**) succeeded in Phase IIa of the clinical trials, results obtained in Phase IIb of the clinical trials stopped the development of this new class of anti-HIV drug, principally because of the natural polymorphism of the CA-SP1 junction that led to a natural resistance of the virus to maturation inhibitors. The prepared pioneer bevirimat-based compounds (**4**–**8**; Figure 1) represented novel, attractive, and promising agents for the future development of the next generation of HIV-1 maturation inhibitors [43]. Their anti-HIV-1 effects are summarized in Table 1.

A search for new methods of antiviral therapy has been primarily focused on the use of substances of natural origin. Recently, a synthesis of a novel series of betulinic acid ester derivatives (**9**–**14**)(**a**–**f**) and **15**–**16** (Figure 2) was published [28]. The structures of the novel compounds were established, and the compounds were tested against DNA and RNA viruses, for antiviral activity against several types of viruses, including HSV-1 [28]. Antiviral experiments and monitoring of the time of addition of the active compound confirmed a research hypothesis and showed high antiviral effect of several derivatives against BEV, H1N1, and HSV-1. Compound **10d** exhibited 6-fold more potent activity against HSV-1 (EC_50_ = 17.2 µM) than the reference drug (acyclovir; EC_50_ = 111.1 µM) (Table 2). Compound **13e** possessed the highest selectivity index (SI = 11.8) even when compared with that of acyclovir (SI = 14.0). Overall, all active compounds showed high virus-specific activity, as none of them showed activity in more than one virus. Most of the active compounds were active at the later steps of the replication cycle. This finding resulted in a suggestion of a mode of action during the step of nucleic acids/protein synthesis, assembly, or maturation. The in silico study was in good agreement with the in vitro data, confirming a high affinity of **10d** to HSV-1 DNA polymerase. In addition, all ester and amide derivatives were tested for antiproliferative activity in A549 and MDCK (Madin-Darby canine kidney) cell lines (Table 3). Ester derivatives in the series of (**9**–**14**)(**a**–**f**) and **15**–**16**, glutaric acid amides **11c** and **11d**, and succinic acid amides **9b** and **9c** showed high cytotoxicity values. These findings provided valuable data for further investigation of the active compounds, and for a subsequent betulin derivation to design novel compounds having higher potential to display antiviral activity. The results indicated that natural resources have still been one of the most important sources of priority structures in a search for new drug candidates.

In a continuing search for novel HIV-1 maturation inhibitors, another series of promising compounds were designed and synthesized on the basis of a number of triterpenoid derivatives (**17**–**17s**; Figure 3), namely the compound **18** (also known GSK3640254 or fipravirimat; Figure 3) [46]. Compound **18** exhibited significantly improved antiviral activity toward a range of clinically relevant polymorphic variants with reduced sensitivity toward the second-generation maturation inhibitor (**17s**; also known as GSK3532795 or BMS-955176). The key structural difference between **18** and its earlier developed analogs (**17**–**17s**) is the replacement of the *para*-substituted benzoic acid moiety located at the C-3 position of the triterpenoid skeleton with a cyclohex-3-ene-1-carboxylic acid substituted with a CH_2_F moiety with the given absolute configuration (**18**; Figure 3). The sp^3^ carbon atom at this site of the molecule provided a new vector for structure-activity relationship exploration and resulted in the identification of compounds with improved polymorphic characteristics while preserving the pharmacokinetic properties of the prototype. This structural element provided a new vector for exploring structure-activity relationships. The approach to the design of **18**, the development of a synthetic route, and its preclinical profile were clearly described in detail in the original paper [46]. Compound **18** has completed the Phase IIa of the clinical trials, in which it demonstrated a dose-related reduction in plasma HIV-1 RNA over 7–10 days, and the compound has been advanced into Phase IIb studies [46].

The investigation of the structure-activity relationships of a series of HIV-1 maturation inhibitors based on the compound **17s** (Figure 3) continued by the subsequent incorporation of novel C-17 amine substituents to reduce the overall basicity of the target compounds [47]. A replacement of the amine group on the C-17 side chain present in **17s** with a tertiary alcohol in combination with either a heterocyclic ring system or a cyclohexyl ring substituted with polar groups provided potent wild-type (WT) HIV-1 maturation inhibitors. They also preserved excellent potency against a T332S/V362I/prR41G variant, a laboratory strain that served as a substitute to assess HIV-1 polymorphic virus characteristics [47]. Compound **19** exhibited a broad anti-HIV-1 activity against relevant Gag polymorphic viruses, and displayed the most desirable overall profile in this series of the studied compounds. In pharmacokinetic studies, **19** had low acquittal and exhibited sufficient oral bioavailability in rats and dogs.

Compounds **18** and **19** had the most desirable overall profile in this series and were evaluated in rat and dog pharmacokinetic studies [47,48]. A comparison of basic anti-HIV-1 activity values is shown in Table 4. An overall summary of antiviral activity values of compounds **17**–**17s**, **18**, and **19** is presented in Table 5 [46,47,48].

Structurally similar compounds to those mentioned above [46,47,48] were reported recently as second-generation maturation inhibitors (compounds **20**–**22** and **23a**–**23e**; Figure 4), displaying effect higher than bevirimat (**3**) against HIV-1 subtype C [49]. In silico studies on the interaction of bevirimat and their analogs have been limited to HIV-1 subtype B (5I4T) due to the lack of an available 3D structure for HIV-1 subtype C virus. The authors [49] have developed a 3D model of the HIV-1C Gag CA-SP1 region using protein homology modeling with HIV-1 subtype B (514T) as a template. The generated HIV-1 C homology model was extensively validated using several tools and served as a template to perform molecular docking studies with eight well-characterized maturation inhibitors. The docked complex of HIV-1C and the studied maturation inhibitors were subjected to molecular dynamics simulation for 100 ns. Based on the obtained data, it was revealed that the investigation was probably a pioneering report on the construction and validation of a 3D model for the HIV-1C Gag CA-SP1, which could serve as a crucial tool in the structure-aided design of novel and well-acting maturation inhibitors [49]. The docking studies confirmed that modifications at the C-28 carbon centers in bevirimat analogs resulted in increased interactions with HIV-1C Gag CA-SP1 and higher binding energy as compared to the parent bevirimat (**3**), which may have conferred antiviral activity to these analogs [49]. The authors [49] presented no antiviral activity data, however, the in silico investigation brought a novel motivation in designing more effective antiviral agents for the next generations. However, antiviral activity data of several compounds of the investigated series can be found in [46].

Phosphate and phosphonate derivatives of betulin (**1**), betulinic acid (**2**) and bevirimat (**3**) represent other types of antiviral compounds displaying better pharmacological characteristics than the parent compounds [1,50]. Thus, several compounds of that series (**24a**–**24h**) are shown in Figure 5, and their antiviral activity values are summarized in Table 6. The inhibitory effect of **24a** (Figure 5, Table 6) was of high value, as well as showed high therapeutic index (IC_50_ = 0.02 µM, TI = 1250) on viral replication, and it displayed high selectivity [1]. The capsid protein (CA) CTD-SP1 might be the target of **24a** against HIV. Among additional phosphate and phosphonate derivatives of bevirimat (**24b**–**24h**), compound **24e** showed antiviral activity comparable with that of **24a**, however, with a slightly worse therapeutic profile than displayed by **24a**.

A novel compound **25** (Figure 6), in principle, also derived from bevirimat (**3**), bearing a pyrazolone system in the molecule, has been considered to represent a HIV-1 maturation inhibitor of the third generation [51]. It displayed a maturation inhibition effect in HIV-1 with the EC_50_ = 20.36 ± 2.85 nM [51]. The mechanism of action of **25** is identical to that of the first-generation antiviral maturation inhibitor bevirimat (**3**). However, the investigation showed that **25** displayed better antiviral potential than bevirimat (**3**) among the virus strains tested, regardless of the presence or absence of human serum [51]. Further designing and developing of suitable molecules resulted in a synthesis of a compound bearing selected, often called “privileged”, structural motifs (Figure 6) [52]. So far the most successful structure **26** (Figure 6) showed high antiviral activity in HIV-1 NL4-3 (EC_50_ = 0.012 µM), which is higher than the antiviral activity of **25**. Within that series of novel piperazine-based compounds (**27a**–**27g**; Figure 6), none of them showed a better antiviral effect and therapeutic profile than **26** (Table 7).

### 2.2. Peptide Derivatives of Triterpenoids of the Lupane, Ursane and Oleanane Family

To demonstrate variability in designing and developing effective structural modifications of bevirimat (**3**), a specific series of peptide analogs of betulinic acid (**2**) should be mentioned. This part of the story starts with triterpenoid saponins, natural products bearing glycoside units, which are a major group of active compounds of natural origin with nonspecific antiviral activities. In turn, the T20 peptide (enfuvirtide), containing a helix zone-binding domain, is a gp41-specific HIV-1 fusion inhibitor. One of the early approaches to the design, synthesis, and structure-activity relationship study of a group of hybrid molecules, in which bioactive triterpenoid sapogenins were covalently bound to the peptides containing the helix zone-binding domain, by the 1,3-dipolar cycloaddition, often known as the click chemistry tool. Thus, a series of triterpenoid-peptide conjugates **28a**–**28n** (Figure 7; Table 8) appeared as early as a decade ago [53]. The investigation resulted in a finding that either the triterpenoid or the peptide part separately showed only weak activity against HIV-1 Env-mediated cell-cell fusion, while the generated hybrid conjugates displayed strong cooperative effects [53]. Among them, P26-BApc (**28k**) exhibited anti-HIV-1 activity against both T20-sensitive and T20-resistant HIV-1 strains and improved pharmacokinetic properties. The results have proven that this scaffold design has been a promising strategy for developing novel HIV-1 fusion inhibitors, and possibly, has encouraged designing novel antiviral therapeutics against other viruses with class I fusion proteins (Table 9) [53].

### 2.3. Miscellaneous Plant Triterpenoids

Investigation of other plant triterpenoids than those of the lupane family can be found in the literature, nevertheless, it is less frequent than could be expected, even if different plant triterpenoids have been reported to display antiviral effects [33].

Ursane-type triterpenoids and 28-nortriterpenoids **29a**−**29i** (Figure 8) were isolated from *Rhododendron latoucheae* [54]. A hyphenated NMR technique (analytical HPLC (high performance liquid chromatography) with a DAD (diode array detector) connected to MS (mass spectrometer), SPE (solid phase extraction), and NMR) has proven effective for the full structural analysis and identification of isolated natural products in complex mixtures [54]. Compounds **29a** and **29i** inhibited HSV-1 in Vero cells with IC_50_ values of 6.4 μM and 0.4 μM, respectively, while the compounds **29b**–**29h** were less effective (Table 10) [54].

The chalcone derivatives of 20-oxo-lupanes have been synthesized and screened for several types of biological activity by Russian authors [55]. Investigating the antiviral activity of the prepared series of compounds, two of them, **30a** and **30b** (Figure 9), were evaluated as compounds displaying anti-HSV-1 activity (Table 11). The antiviral activity of **30a** and **30b** was tested against HCMV, and that of **30b** also against HSV-1 and HPV.

A large series of new pentacyclic triterpenoids, including oleanane-type, ursane-type, and taraxerane-type, were isolated from the stems and branches of *Enkianthus chinensis* [56]. Their structures were elucidated by extensive spectroscopic analyses, X-ray crystallographic data, and electronic circular dichroism (ECD) techniques. The in vitro biological activity evaluation resulted in a finding that the three compounds (**31a**–**31c**; Figure 10) showed antiviral activity. The most active compound of this small series of natural triterpenoids was **31c** showing latent activity against HSV-1 with an IC_50_ value of 6.4 μM. Their structures and antiviral activity values (in comparison with those of acyclovir used as the positive reference agent) are summarized in Table 12.

Even if triterpenoids and their natural saponin derivatives exhibit anti-HSV-1 activity, there has still been a lack of comprehensive information on the anti-HSV-1 activity of triterpenoids. Therefore, expanding information on the anti-HSV-1 activity of triterpenes and improving the efficiency of their exploration are urgently required. To improve the efficiency of the development of anti-HSV-1 active compounds, recently, Japanese authors constructed a predictive model for the anti-HSV-1 activity of triterpenes by using the information obtained from previous studies [57]. They constructed a binary classification model (i.e., active or inactive) using a logistic regression algorithm. As a result, the assay was performed on 20 triterpenes and triterpenoids, finally identifying the structure **32** (Figure 11) as a potent anti-HSV-1 compound, displaying IC_50_ = 13.06 μM.

Several so far undescribed cycloartane triterpenoids, pseudolarnoids A−G, together with other known triterpenoids, were isolated from the seeds of *Pseudolarix amabilis* (J. Nelson) Rehder [58]. Their structures were elucidated on the basis of spectroscopic analysis, X-ray crystallography, and ECD data. Three of these natural products (**33**–**35**; Figure 12) proved their ability to display potent antiviral effects on HSV-1 in vitro (Table 13). Based on the therapeutic index values, structures **34** and **35** showed a better therapeutic profile than **33** (Table 13). The structures of other less active or inactive cycloartane triterpenoids are not shown, they can be found in the original paper [58].

## 3. Conclusions

Even if the presented review paper covers a short period of investigation of antiviral agents, it shows a wide variety of triterpenoid-based compounds investigated as potential antiviral agents. Even if a majority of structures are—in principle—structurally derived from bevirimat (**3**), other types of triterpenoid structures were also included in this type of investigation. Because studies made with the lupane family of triterpenoids were always performed with high intensity, and possibly with high priority as well, the results achieved so far reveal that lupane-based agents seem to be the most successful structures among all triterpenoid-based ones. The review shows achievements in searching for novel structures and clearly shows that this intensive investigation is resulting in designing perspective structures that have a great chance to pass over all phases of clinical trials to give a potent antiviral agent for application in human medicine.

Based on the literature search made while collecting the relevant papers for this review paper, a conclusion was made that no one of the triterpenoids, i.e., other than lupane, when used as a source for designing and developing highly potent antiviral agents, resulted in achieving active compounds with a practical impact. Our current investigation has revealed a very preliminary finding supporting the opposite opinion. Those results will be published later when the relevant data is collected.

## Figures and Tables

**Figure 1 molecules-28-07718-f001:**
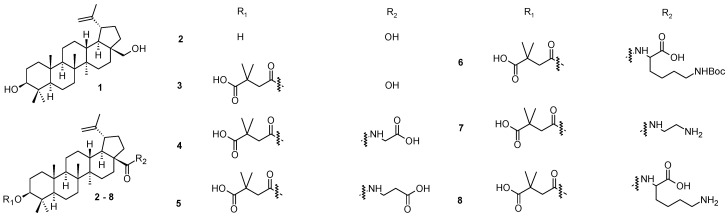
Betulin (**1**), betulinic acid (**2**), bevirimat (**3**) and its early stage analogs (**4**–**8**).

**Figure 2 molecules-28-07718-f002:**
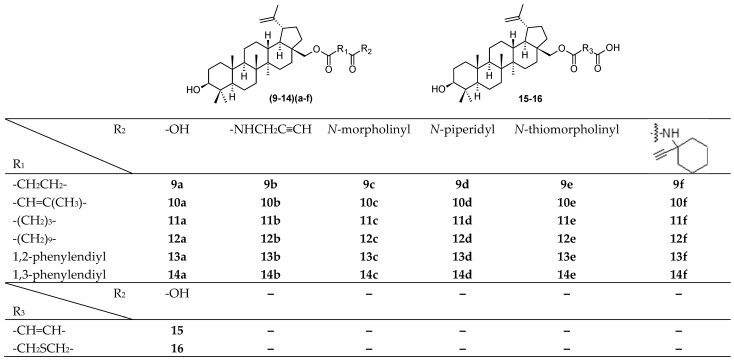
Structures of compounds (**9**–**14**)(**a**–**f**) and **15**–**16**.

**Figure 3 molecules-28-07718-f003:**
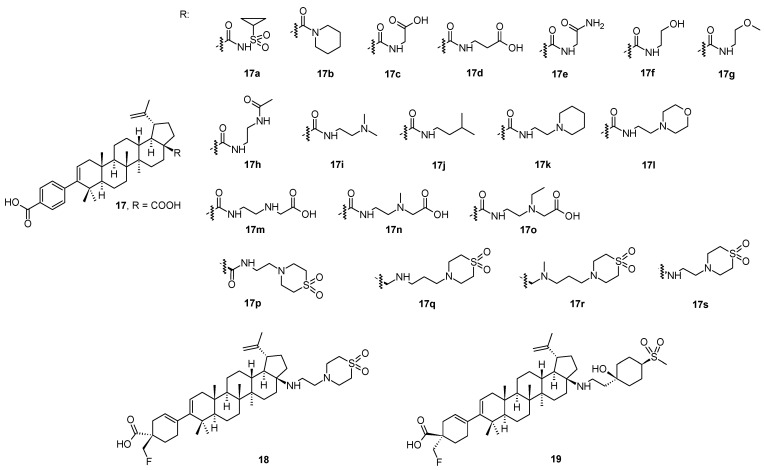
Structures of the compounds **17**–**17s**, **18** and **19**.

**Figure 4 molecules-28-07718-f004:**
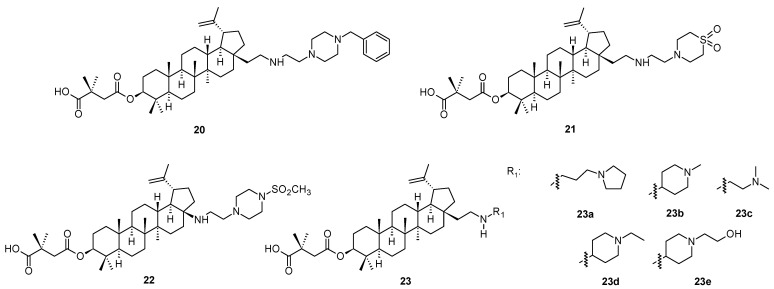
Structures of compounds **20**–**22** and **23a**–**23e**.

**Figure 5 molecules-28-07718-f005:**
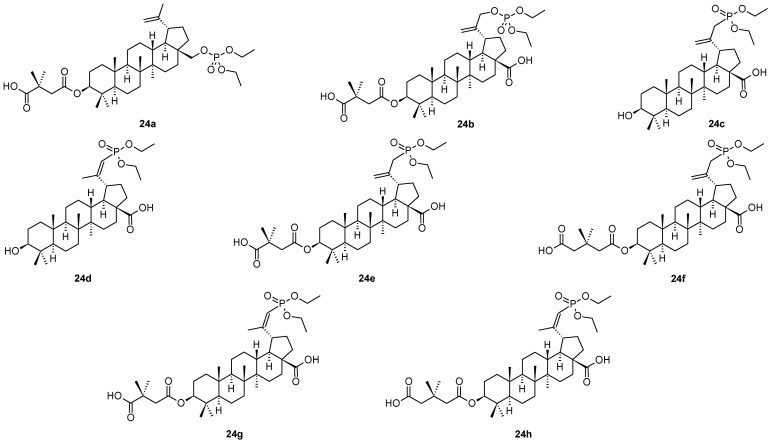
Structures of compounds **24a**–**24h**.

**Figure 6 molecules-28-07718-f006:**
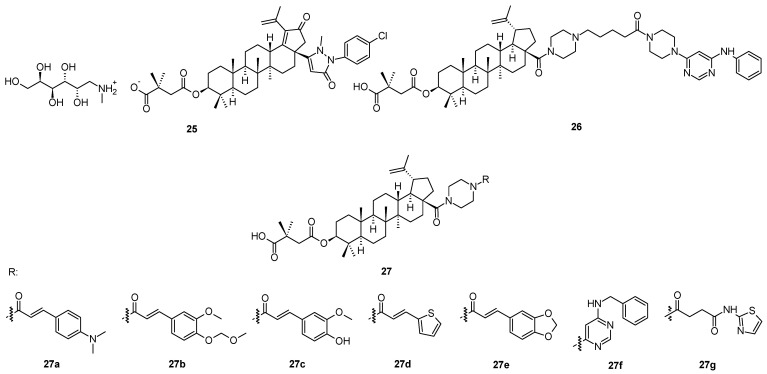
Structures of compounds **25**, **26** and **27a**–**27g**.

**Figure 7 molecules-28-07718-f007:**
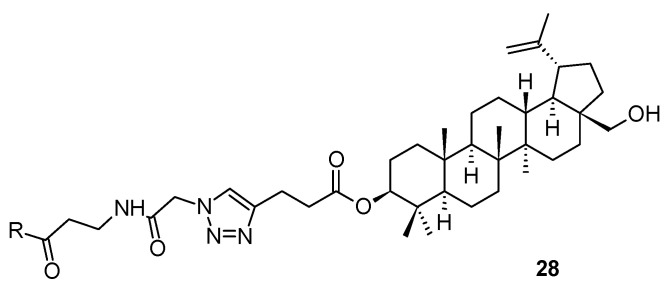
Triterpenoid-peptide hybrid conjugates. For explanation of the substituent R, see Table 8.

**Figure 8 molecules-28-07718-f008:**
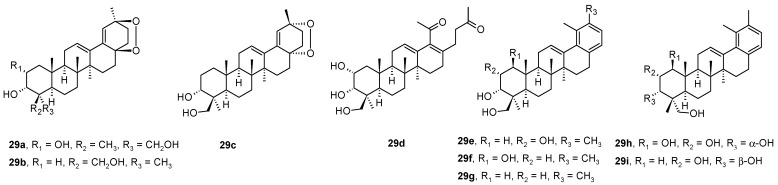
Structures of the compounds **29a**–**29i**.

**Figure 9 molecules-28-07718-f009:**
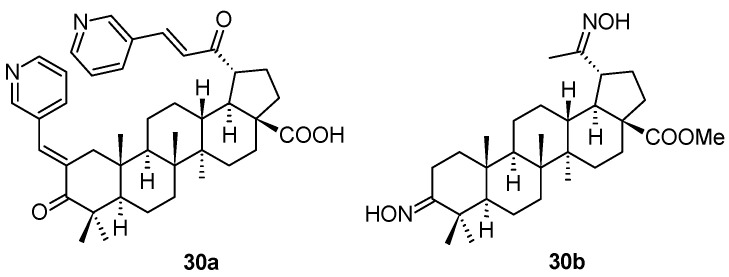
Structures of the compounds **30a** and **30b**.

**Figure 10 molecules-28-07718-f010:**
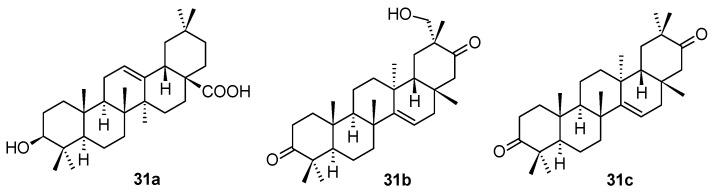
Structures of the compounds **31a**–**31c**.

**Figure 11 molecules-28-07718-f011:**
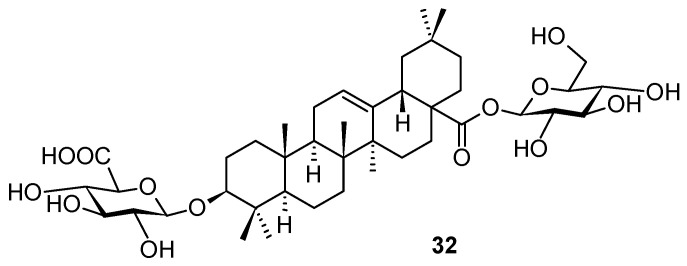
Structure of the natural compound **32**.

**Figure 12 molecules-28-07718-f012:**
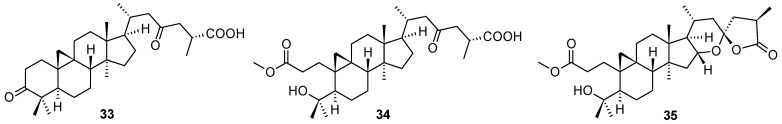
Structures of the natural compounds **33**–**35**.

**Table 1 molecules-28-07718-t001:** Efficiency of HIV-1 infection inhibition by betulinic acid derivatives [43].

Compd.	IC_50_ [µM]	CC_50_ [µM]	SI ^a^
**2**	5.315	4.52	0.85
**3**	0.040	31.00	775.00
**4**	0.160	49.50	309.37
**5**	0.118	48.50	411.01
**6**	0.170	33.90	199.41
**7**	0.016	33.90	2118.75
**8**	4.330	48.70	11.24

^a^ The selectivity index (SI) represented the CC_50_/IC_50_ ratio.

**Table 2 molecules-28-07718-t002:** Anti-HSV-1 activity of the studied compounds in the mode III post-treatment assay. Compounds that reduced virus titers by at least 2 logarithms were considered active and are presented here. Presented values are medians from three independent experiments [28].

Compd.	CC_50_ [μM]	EC_50_ [μM]	SI	Compd.	CC_50_ [μM]	EC_50_ [μM]	SI
**10d**	68.7	17.2	4.0	**13c**	143.5	35.9	4.0
**11a**	80.8	22.4	3.6	**13d**	65.4	32.7	2.0
**11c**	150.8	18.9	8.0	**13e**	349.6	29.6	11.8
**12c**	164.1	34.8	4.7	Acyclovir	1555.6	111.1	14.0

**Table 3 molecules-28-07718-t003:** In vitro cytotoxicity of tested compounds **(9**–**14)(a**–**f) and 15**–**16** on human lung cancer cell line (A549) and normal canine kidney cell line (MDCK) [28].

Compd.	CC_20_ [µM] ^a,b^		Compd.	CC_20_ [µM] ^a,b^	
	A549	MDCK		A549	MDCK
**1**	28.22	28.22	**12b**	38.46	153.85
**9a**	11.51	11.51	**12c**	34.77	139.08
**9b**	10.78	5.34	**12d**	127.23	127.84
**9c**	9.63	4.78	**12e**	114.81	1141.81
**9d**	69.83	17.46	**12f**	102.46	102.46
**9e**	69.42	124.84	**13a**	21.15	42.30
**9f**	110.38	0.77	**13b**	39.81	159.24
**10a**	11.26	5.63	**13c**	35.87	143.47
**10b**	10.56	84.46	**13d**	65.45	32.72
**10c**	37.82	37.82	**13e**	58.89	7.30
**10d**	34.34	4.26	**13f**	104.82	1.57
**10e**	61.50	61.50	**14a**	21.15	42.30
**10f**	54.47	54.47	**14b**	159.24	159.24
**11a**	22.44	11.22	**14c**	71.74	143.47
**11b**	42.09	1.18	**14d**	130.89	130.89
**11c**	18.85	18.85	**14e**	117.79	117.79
**11d**	17.12	4.25	**14f**	104.82	104.82
**11e**	122.70	61.35	**15**	11.55	5.73
**11f**	108.70	108.70	**16**	21.08	21.08
**12a**	5.09	10.20	Acyclovir	196.08	NT ^c^

^a^ Presented values are medians from three independent experiments; ^b^ CC_20_, concentrations required to reduce A549 and MDCK cells viability by 20%; ^c^ NT = not tested.

**Table 4 molecules-28-07718-t004:** A comparison of anti-HIV-1 activity values of bevirimat (**3**) and the compounds **17**–**19** [47,48].

3	17	18	19
WT, EC_50_ = 10.0 nM	WT, EC_50_ = 5.0 nM	WT, EC_50_ = 3.0 nM	WT, EC_50_ = 1.6 nM
40% HS, EC_50_ = 970.0 nM (97×)	40% HS, EC_50_ = 10.0 nM (2×)	40% HS, EC_50_ = 14.0 nM (4.6×)	40% HS, EC_50_ = 9.3 nM (5.8×)
V370A, EC_50_ = 552.0 nM	V370A, EC_50_ = 6.0 nM	V370A, EC_50_ = 2.0 nM	V370A, EC_50_ = 3.0 nM
ΔV370, EC_50_ ≥ 10,000 nM	ΔV370, EC_50_ = 6.0 nM	ΔV370, EC_50_ = 3.6 nM	ΔV370, EC_50_ = 5.1 nM
	T332S/V362I/prR41G, EC_50_ = 704 nM	T332S/V362I/prR41G, EC_50_ = 7.0 nM	T332S/V362I/prR41G, EC_50_ = 6.4 nM

**Table 5 molecules-28-07718-t005:** Anti-HIV-1 activity of compounds **17**–**17s**, **18** and **19** [46,47,48].

Compd.	EC_50_ [nM]	EC_50_ V370A [nM]	EC_50_ ΔV370 [nM]	Compd.	EC_50_ [nM]	EC_50_ V370A [nM]	EC_50_ ΔV370 [nM]
**17**	16	233	>3000	**17k**	7	10	359
**17a**	59	67	–	**17l**	7	12	77
**17b**	369	NT	–	**17m**	42	15	61
**17c**	106	>2000	–	**17n**	10	51	160
**17d**	37	427	>4000	**17o**	3	9	64
**17e**	6	32	2000	**17p**	2	6	60
**17f**	7	41	>2000	**17q**	1	2	13
**17g**	17	150	–	**17r**	1	2	13
**17h**	5	24	361	**17s**	2	3	13
**17i**	3	8	31	**18**	3	2	3.6
**17j**	47	101	1600	**19**	1.6	3.0	5.1

**Table 6 molecules-28-07718-t006:** Anti-HIV-1 activity (EC_50_ [µM]) of compounds **24a**–**24h** [1,50].

Compd.	24a	24b	24c	24d	24e	24f	24g	24h
EC_50_ [µM]	0.02 ^a^	1	>10	>10	0.02	0.9	4	0.6

^a^ Therapeutic index, TI = 1250.

**Table 7 molecules-28-07718-t007:** Antiviral activity of the piperazine-type compounds **27a**–**27g**, compared to **25** and **26** in HIV-1 NL4-3 [51,52].

Compd.	25	26	27a	27b	27c	27d	27e	27f	27g
EC_50_ [µM]	0.021	0.012	0.027	0.025	0.018	0.032	0.022	0.037	0.040

**Table 8 molecules-28-07718-t008:** Inhibitory activities of sapogenin-peptide conjugates **28a**–**28n** on HIV-1 Env-mediated cell-cell fusion [53].

Compd.	Compd. Code ^a^	Sequence ^b^	EC_50_ [nM] ^c^
**28a**	BAo—P26	BAo-a-NNYTSLIHSLIEESQNQQEKNEQELL	89.4 ± 2.4
**28b**	UAo—P26	UAo-a-NNYTSLIHSLIEESQNQQEKNEQELL	145 ± 17
**28c**	OAo—P26	OAo-a-NNYTSLIHSLIEESQNQQEKNEQELL	176 ± 45
**28d**	BAc—P26	BAc-a-NNYTSLIHSLIEESQNQQEKNEQELL	15.1 ± 2.5
**28e**	UAc—P26	UAc-a-NNYTSLIHSLIEESQNQQEKNEQELL	51.5 ± 25
**28f**	OAc—P26	OAc-a-NNYTSLIHSLIEESQNQQEKNEQELL	28.6 ± 5.1
**28g**	BApc—P26	BApc-a-NNYTSLIHSLIEESQNQQEKNEQELL	197 ± 55
**28h**	BApo—P26	BApo-a-NNYTSLIHSLIEESQNQQEKNEQELL	327 ± 21
**28i**	P26—BAo	NNYTSLIHSLIEESQNQQEKNEQELL-a-K(BAo)	19.6 ± 5.0
**28j**	P26−BAc	NNYTSLIHSLIEESQNQQEKNEQELL-a-K(BAc)	44.2 ± 10
**28k**	P26 − BApc	NNYTSLIHSLIEESQNQQEKNEQELL-a-K(BApc)	3.94 ± 0.3
**28l**	P26 − BApo	NNYTSLIHSLIEESQNQQEKNEQELL-a-K(BApo)	7.94 ± 1.5
**28m**	P26 − UApc	NNYTSLIHSLIEESQNQQEKNEQELL-a-K(UApc)	3.35 ± 1.1
**28n**	P26 − OApc	NNYTSLIHSLIEESQNQQEKNEQELL-a-K(OApc)	3.31 ± 1.0
**2**	BA	Betulinic acid	>1,000,000
**–**	UA	Ursolic acid	>1,000,000
**–**	OA	Oleanolic acid	>1,000,000
**–**	P26	NNYTSLIHSLIEESQNQQEKNEQELL	3240 ± 560
**–**	Ptrz—P26	Ptrz-a-NNYTSLIHSLIEESQNQQEKNEQELL	3580 ± 156
**–**	P26—Ptrz	NNYTSLIHSLIEESQNQQEKNEQELL-a-K(Ptrz)	2183 ± 786
**–**	P26 + BAo	NNYTSLIHSLIEESQNQQEKNEQELL + BAo	2390 ± 612
**–**	T20	YTSLIHSLIEESQNQQEKNEQELLELDKWASLWNWF	10.1 ± 1.4

^a^ When a non-peptide moiety is conjugated to the *N*-terminus of P26, the hybrid has carboxyamide at the C-terminus. When a non-peptide moiety is attached to the C-terminus of P26, the conjugate has an acetyl group at the *N*-terminus and carboxyamide at the C-terminus. P26 and T20 have an acetyl group at the *N*-terminus and carboxyamide at the C-terminus; ^b^ a = β-alanine; Ptrz = 4-propyl-1*H*-1,2,3-triazol; ^c^ Compounds were tested in triplicate, and the data are presented as the mean ± standard deviation (SD).

**Table 9 molecules-28-07718-t009:** Anti-HIV-1 activity and cytotoxicity values of the conjugates **28a**–**28n** [53].

Compd.	Compd. Code	EC_50_ [nM] for Inhibition ^a^			
		HIV-1_IIIB_ Replication	HIV-1_BaL_ Replication	CC_50_ [μM]	SI ^b^
**28a**	BAo—P26	475 ± 87	456 ± 72	>25	>52
**28b**	UAo—P26	565 ± 130	288 ± 32	>25	>44
**28c**	OAo—P26	369 ± 2.0	498 ± 87	>25	>68
**28d**	BAc—P26	133 ± 53	98.0 ± 26	>25	>187
**28e**	UAc—P26	387 ± 252	113 ± 55	>25	>64
**28f**	OAc—P26	94.0 ± 15	150 ± 21	>25	>266
**28g**	BApc—P26	242 ± 11	519 ± 98	>25	>103
**28h**	BApo—P26	501 ± 122	99.0 ± 12	>25	>52
**28i**	P26—BAo	154 ± 9.0	135 ± 15	>25	>50
**28j**	P26—BAc	61.6 ± 16	83.1 ± 8.3	>25	>406
**28k**	P26—BApc	4.28 ± 0.7	6.90 ± 0.1	14.3 ± 1.0	3348
**28l**	P26—BApo	475 ± 87	456 ± 72	>25	>52
**28m**	P26—UApc	565 ± 130	288 ± 32	>25	>44
**28n**	P26—OApc	369 ± 2.0	498 ± 87	>25	>68

^a^ Compounds were tested in triplicate, and the data are presented as the mean ± standard deviation; ^b^ SI (selectivity index) = CC_50_/EC_50_ for inhibiting HIV-1_IIIB_ infection.

**Table 10 molecules-28-07718-t010:** Antiviral activity against HSV-1 and cytotoxicity for compounds **29a**–**29i** in Vero cells ^a^ [54].

Compd.	CC_50_ [µM] ^b^	IC_50_ [µM]	SI ^c^
**29a**	23.11	6.41	3.6
**29b**	57.74	>11.11	-
**29c**	33.33	>11.11	-
**29d**	69.34	>33.33	-
**29e**	3.70	>1.23	-
**29f**	11.11	>3.70	-
**29g**	5.34	>1.23	-
**29h**	1.78	>0.41	-
**29i**	1.78	0.41	4.3
Acyclovir	>100	0.41	>243.9

^a^ Data represent mean values for three independent determinations; ^b^ Cytotoxic concentration required to inhibit Vero cell growth by 50%; ^c^ Selectivity index value equaled CC_50_/IC_50_.

**Table 11 molecules-28-07718-t011:** Antiviral activity values found for **30a** and **30b** [55].

Compd.	EC_50_ [μM]	Compd.	EC_50_ [μM]
**30a**	>0.24 (HCMV)	**30b**	1.20 (HSV-1)
			3.47 (HPV)

**Table 12 molecules-28-07718-t012:** Antiviral activities of **31a**–**31c** against HSV-1 in Vero cells [56].

Compd.	CC_50_ [μM]	IC_50_ [μM]	SI ^a^
**31a**	33.3	11.1	3.0
**31b**	57.7	14.3	4.0
**31c**	57.7	6.4	9.0
Acyclovir	>100	0.3	>370.4

^a^ Selectivity index value calculated as a ratio CC_50_/IC_50_.

**Table 13 molecules-28-07718-t013:** Antiviral activity values found for **33**–**35** against HSV-1 [58].

Compd.	IC_50_ [μM]	SI ^a^
**33**	15.3 ± 1.9	2.4 ± 0.3
**34**	1.1 ± 0.2	6.8 ± 0.9
**35**	4.3 ± 0.4	7.8 ± 0.7
Acyclovir	11.9 ± 1.4	>50

^a^ Selectivity index value calculated as a ratio CC_50_/IC_50_; CC_50_ values were calculated in silico.

## Data Availability

Not applicable.

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
