# Peer review of "Selected Plant Triterpenoids and Their Derivatives as Antiviral Agents"

_molecules, 2023, doi:10.3390/molecules28237718_

Round 1

Reviewer 1 Report

Comments and Suggestions for Authors

This review describes the results of the most recent investigation of triterpenoid-based antiviral agents effective namely in the HIV-1 and HSV-1 treatment.

This review is a summary of results related to this derivative, and I think many readers will be interested. However, there are some mistakes, so please consider correcting them.

The corrections are shown below.

1.     Figure 1: should be corrected “R” to “R2”.

2.     Figure 2: should be corrected “9-14” to “R9a-14a”. 

3.     Figure 2: should be corrected “(9-14)(a-e)” to “(9-14)(b-f)”.

4.     Figure 2: The structural formula in the Figure 2 should be one.

5.     "Compound" in all Tables should be unified to "Compd."

6.     Figure 6: should be added anO” atom to the pyrazolone moiety in compound 25.

7.     Table 12: Please corrected compound number “31a, 31b, and 31c” to bold.

Author Response

Reviewer 1:

Comments and Suggestions for Authors:

This review describes the results of the most recent investigation of triterpenoid-based antiviral agents effective namely in the HIV-1 and HSV-1 treatment.

This review is a summary of results related to this derivative, and I think many readers will be interested. However, there are some mistakes, so please consider correcting them.

The corrections are shown below.

  1. Figure 1: should be corrected “R” to “R2”.

Answer: The substituent was modified accordingly.

  1. Figure 2: should be corrected “9-14” to “R9a-14a”. 

Answer: The Figure 2 was modified for better understanding.

  1. Figure 2: should be corrected “(9-14)(a-e)” to “(9-14)(b-f)”.

Answer: The Figure 2 was modified for better understanding.

  1. Figure 2: The structural formula in the Figure 2 should be one.

Answer: The Figure 2 was modified for better understanding.

  1. "Compound" in all Tables should be unified to "Compd."

Answer: The word was modified in all Tables accordingly.

  1. Figure 6: should be added an“O” atom to the pyrazolone moiety in compound 25.

Answer: The missing oxygen atom in the compound 25 was added to the structure.

  1. Table 12: Please corrected compound number “31a, 31b, and 31c” to bold.

Answer: The numbers of the compounds were corrected to “bold”.

Reviewer 2 Report

Comments and Suggestions for Authors

Molecules-2698812_2023 The manuscript entitled “Selected Plant Triterpenoids and Their Derivatives
as Antiviral Agents” by Martina Wimmerová, Uladzimir Bildziukevich, Zdenek
Wimmer submitted to Molecules for ‘Natural Compounds in Plants’, 2nd.

The manuscript provides a review and is devoted to a very actively developing field of  of plant-derived products for the needs of pharmacotherapy,  namely, triterpenoids. In particular, the authors summarized and discuss the antiviral properties of betulin and betulinic acid derivatives, which are natural compounds and by-products of the forest industry.

In recent years, betulin-based compounds have become a research hotspot in the field of phytochemistry.

Currently, attention in this field is mainly focused on their antiviral, antimalarial and anticancer potential.

In general, thanks to significant progress, research into antiviral and other properties is in an active stage.

The authors carried out a thorough literature search on the topic, the results of which seem useful for the development of this area and further progress in the creation of new drugs.

The literature material is well organized. This review cites 58 references, predominantly research articles and reviews.

These materials could be useful to specialists in the field of medicinal chemistry and pharmacology. They will also be of interest to biochemists and chemists working with bioactive compounds.

In general, the manuscript is well prepared, although at times it is presented in too much detail, or rather the materials of the cited references are copied, and at the same time, I lacked an analytical approach in the presented review.

The manuscript does not raise any objections, and can be published in Molecules after minor revision.

Several details and inaccuracies should be noted.

1.    Lines 141-143. Repeat information. I recommend removing it. Betulin and betulinic acid have already been discussed earlier (see lines 93-100).

2.    Figure 2. Figure 2 actually represents both a figure and a table. In order not to mislead the reader, it is better to present the material as Fig. with formulas and in table form. The first two formulas do not differ from each other.

3.    Further, N-morpholinyl and N-piperidyl, etc. are more clearly represented in the table by formulas. Moreover, compounds 10c and 13d, the effects of which are discussed in the text, are better represented by formulas, but already in the figure.

4.    Lines 277-278. In my opinion, there is no relationship with bevirimat and peptide derivatives.

5.    Tables. The heading of the Tables should indicate: data from [ref.]. This recommendation applies to all tables.

6.    The conclusion should be edited to reflect the main trends in the development of plant triterpenoid derivatives as antiviral agents.

Comments on the Quality of English Language

English is ok.

Author Response

Reviewer 2:

Comments and Suggestions for Authors:

Molecules-2698812_2023 The manuscript entitled “Selected Plant Triterpenoids and Their Derivatives as Antiviral Agents” by Martina Wimmerová, Uladzimir Bildziukevich, Zdenek
Wimmer submitted to Molecules for ‘Natural Compounds in Plants’, 2nd.

The manuscript provides a review and is devoted to a very actively developing field of  of plant-derived products for the needs of pharmacotherapy,  namely, triterpenoids. In particular, the authors summarized and discuss the antiviral properties of betulin and betulinic acid derivatives, which are natural compounds and by-products of the forest industry.

In recent years, betulin-based compounds have become a research hotspot in the field of phytochemistry.

Currently, attention in this field is mainly focused on their antiviral, antimalarial and anticancer potential.

In general, thanks to significant progress, research into antiviral and other properties is in an active stage.

The authors carried out a thorough literature search on the topic, the results of which seem useful for the development of this area and further progress in the creation of new drugs.

The literature material is well organized. This review cites 58 references, predominantly research articles and reviews.

These materials could be useful to specialists in the field of medicinal chemistry and pharmacology. They will also be of interest to biochemists and chemists working with bioactive compounds.

In general, the manuscript is well prepared, although at times it is presented in too much detail, or rather the materials of the cited references are copied, and at the same time, I lacked an analytical approach in the presented review.

The manuscript does not raise any objections, and can be published in Molecules after minor revision.

Several details and inaccuracies should be noted.

  1. Lines 141-143. Repeat information. I recommend removing it. Betulin and betulinic acid have already been discussed earlier (see lines 93-100).

Answer: The repeating items of information were deleted.

  1. Figure 2. Figure 2 actually represents both a figure and a table. In order not to mislead the reader, it is better to present the material as Fig. with formulas and in table form. The first two formulas do not differ from each other.

Answer: The Figure 2 was modified for better understanding.

  1. Further, N-morpholinyl and N-piperidyl, etc. are more clearly represented in the table by formulas. Moreover, compounds 10c and 13d, the effects of which are discussed in the text, are better represented by formulas, but already in the figure.

Answer: The substituents as N-morpholinyl, N-piperidyl and N-thiomorpholinyl seem to be so simple that we believe it is not necessary to draw their structures that may be found in the original literature. The compounds 10c (now 10d) and 13d (now 13e) are sufficiently characterized by substituents to be easily introduced to their general formulae in Figure 2.

  1. Lines 277-278. In my opinion, there is no relationship with bevirimat and peptide derivatives.

Answer: There is no relationship in the structures, indeed, however, there is a relationship in the antiviral activity of the peptide derivatives with bevirimat. Including the peptide derivatives into the review paper should demonstrate that even structurally different compound may display antiviral effect comparable with that of bevirimat.

  1. Tables. The heading of the Tables should indicate: data from [ref.]. This recommendation applies to all tables.

Answer: The required data – reference numbers – were added to the headings of all Tables.

  1. The conclusion should be edited to reflect the main trends in the development of plant triterpenoid derivatives as antiviral agents.

Answer: The conclusion was modified and extended.

Round 2

Reviewer 1 Report

Comments and Suggestions for Authors

I reviewed the revised manuscript.

The manuscript has been revised well.

I agree with accepting this manuscript.